# Zinc Oxide Nanoparticles Exhibit Favorable Properties to Promote Tissue Integration of Biomaterials

**DOI:** 10.3390/biomedicines9101462

**Published:** 2021-10-13

**Authors:** Nadine Wiesmann, Simone Mendler, Christoph R. Buhr, Ulrike Ritz, Peer W. Kämmerer, Juergen Brieger

**Affiliations:** 1Department of Otorhinolaryngology, University Medical Center Mainz, Langenbeckstrasse 1, 55131 Mainz, Germany; mendler@uni-mainz.de (S.M.); christoph.raphael.buhr@gmail.com (C.R.B.); brieger@uni-mainz.de (J.B.); 2Department of Oral- and Maxillofacial Surgery, University Medical Center Mainz, Augustusplatz 2, 55131 Mainz, Germany; peer.kaemmerer@unimedizin-mainz.de; 3Department of Orthopedics and Traumatology, University Medical Center Mainz, Langenbeckstrasse 1, 55131 Mainz, Germany; ulrike.ritz@unimedizin-mainz.de

**Keywords:** zinc oxide nanoparticles, biomaterials, implants, angiogenesis, healing process, antibacterial, fibroblasts, endothelial cells, CAM assay

## Abstract

Due to the demographic change, medicine faces a growing demand for tissue engineering solutions and implants. Often, satisfying tissue regeneration is difficult to achieve especially when co-morbidities hamper the healing process. As a novel strategy, we propose the incorporation of zinc oxide nanoparticles (ZnO NPs) into biomaterials to improve tissue regeneration. Due to their wide range of biocompatibility and their antibacterial properties, ZnO NPs are already discussed for different medical applications. As there are versatile possibilities of modifying their form, size, and function, they are becoming increasingly attractive for tissue engineering. In our study, in addition to antibacterial effects of ZnO NPs, we show for the first time that ZnO NPs can foster the metabolic activity of fibroblasts as well as endothelial cells, both cell types being crucial for successful implant integration. With the gelatin sponge method performed on the chicken embryo’s chorioallantoic membrane (CAM), we furthermore confirmed the high biocompatibility of ZnO NPs. In summary, we found ZnO NPs to have very favorable properties for the modification of biomaterials. Here, incorporation of ZnO NPs could help to guide the tissue reaction and promote complication-free healing.

## 1. Introduction

In modern medicine, the demand for various biomaterials, such as bone or soft tissue replacements, is increasing continuously. The requirements for their stability, size, and shape are well defined and depend on their intended function. Furthermore, there is a desire for the material to be able to interact with the adjacent tissue to achieve wound healing and tissue integration. Additionally, long-term stability of the biomaterial must be ensured to achieve successful function and regeneration. It has been known for a long time that a material’s surface and structure are of pivotal importance for ensuring good tissue integration [1,2,3,4,5,6], but also, different sites and stability issues must be addressed. In this context, nanotechnology has opened new possibilities for actively designing the desired tissue reaction. Moreover, unfavorable tissue reactions to the foreign material and the risk of infection of the transplanted or implanted biomaterial are still major issues. One common complication after implant insertion is an infection in the initial phase after implantation, which normally must be fought with systemic antibiotic treatment [7]. In this context, it has often proven to be difficult to reach the local site of infection with a systemic antibiotic concentration high enough to stop the respective biofilm formation [8]. Furthermore, bacterial antibiotic resistances are on the rise, and there is demand for new ways to contain bacterial growth and prevent infection [9].

Thus, it would be advantageous to have a substance directly at the site of implantation that hampers bacterial infection and furthers wound healing. As such a substance, we propose zinc oxide nanoparticles (ZnO NPs). Their antibacterial properties are well known and characterized [10,11,12,13]. ZnO NPs exhibit a good biocompatibility [14], and they can be degraded over time by the body such that an excess of zinc can be excreted [15]. In contrast with antibiotics, ZnO NPs do not promote the development of bacterial strains with antibiotic resistances [16], which is a serious global healthcare problem [9]. Therefore, ZnO NPs currently come into focus for different tissue engineering and wound healing applications [17,18,19].

First studies were able to show that incorporation of a small percentage of ZnO NPs into a biomaterial can also help to promote tissue integration of the material by fostering the adherence of fibroblasts, neo-angiogenesis, and wound healing [20,21,22,23]. It is suspected that ZnO NPs can promote the upregulation of pro-angiogenic factors such as the vascular endothelial growth factor (VEGF) by the generation of reactive oxygen species (ROS) [23]. We could show that ZnO NPs can release zinc ions [24] and induce the generation of ROS [25]. Thereby they can interact with their surrounding tissue via multiple mechanisms.

Wound healing at the cellular level is a complex process orchestrated by different cell populations. Irrespective of the type of wounded tissue, tissue repair follows four phases: coagulation and hemostasis, inflammation, proliferation, and remodeling [26]. The body’s response to foreign material can be considered as a modified variant of wound healing, and fibroblasts are key participants in this process [27]. In the scope of tissue repair, resident fibroblasts begin to proliferate and migrate into the fibrin clot that was formed. By deposition of extracellular matrix, contraction of the wound, and breakdown of the fibrin clot, they shift the wound microenvironment from the inflammatory state to the growth state [28,29]. Furthermore, an effective angiogenic response is a crucial event for both wound healing in general and biomaterial integration in particular. Angiogenesis ensures the supply of adequate amounts of nutrients and oxygen, it helps to eliminate waste products along with carbon dioxide, and it is involved in the recruitment of the cells necessary for the healing process [30].

We propose that a combination of antibacterial properties and the promotion of fibroblasts and endothelial cells could positively influence the prevention of post-implantation infections and the promotion of tissue integration of the implant material. The high biocompatibility of zinc makes the metal ideal for the supplementation of implant materials, while harboring antibacterial and wound-healing properties. Thus, it could pave the way for fewer complications when inserting implants and transplants into the human body.

While there are many studies showing the integration of zinc oxide into different materials and investigating the properties of the composite material, the effect of ZnO NPs as a single substance on different cell types has not been investigated much. In particular, the effect of low ZnO NPs concentrations on non-tumorigenic cell types has remained largely unstudied. However, in the biomedical field, this is precisely where the great potential lies for the application of ZnO NPs. Because zinc is the second most common metal trace element after iron, it is already present in the body, and it is involved in numerous biological functions.

In the scope of this study, we aimed at investigating the antibacterial properties of ZnO NPs as well as their influence on fibroblasts and endothelial cells in vitro to assess the potential of ZnO NPs to further wound healing. Moreover, the particles’ behavior in vivo on the chorioallantoic membrane was analyzed.

## 2. Materials and Methods

### 2.1. Zinc Oxide Nanoparticles (ZnO NPs)

Zinc oxide nanoparticles were obtained from IoLiTec Ionic Liquids Technologies GmbH (Product Nr.: NO-0011-HP, Heilbronn, Germany). ZnO NPs were produced wet-chemically, they were nearly spherical, and the average particle size was 20 nm, determined by transmission electron microscopy. ZnO NPs were stored airtightly sealed at 8 °C and 1 mg/mL ZnO NPs dispersions were freshly prepared immediately before each experiment with high-purity water (Aqua ad iniectabilia Braun, B. Braun Melsungen AG, Melsungen, Germany). To disperse the NPs, they were sonicated 5 min with 220–240 V and 37 kHz in an Elmasonic S 40 sonicator (Elma Schmidbauer GmbH, Singen, Germany), and sedimentation was prevented by pipetting up and down immediately before usage of the NP dispersion. ZnCl_2_ solution was obtained from Sigma-Aldrich (0.1 M solution, Sigma-Aldrich, St. Louis, MO, USA) and diluted with high-purity water.

### 2.2. Assessment of Antibacterial Activity

*Escherichia coli* bacteria were grown in an overnight culture. After measuring the OD (OD600 = optical density at 600 nm), a starter culture was established in 300 mL LB medium by addition of 700 µL of the overnight culture. The OD of the starter culture was measured again to ensure equal starting conditions in different runs of the experiment. The starter culture was then distributed to Erlenmeyer flasks, which were filled with 20 mL bacterial suspension each for each dosage group. Bacteria were then treated with the indicated amounts of ZnO NPs or ZnCl_2_ and incubated at a horizontal shaker at 140 rpm and 37 °C. The OD of the bacterial suspension was measured at the indicated time points to monitor bacterial growth. Bacteria treated with a corresponding amount of water served as positive control; bacteria treated with 0.25 mg/mL ampicillin served as negative control. The OD of the bacterial suspension at time = 0 h was subtracted from the OD at the following time points to correct for the increasing opacity of the LB medium with increasing NP concentrations.

### 2.3. Cells and Cell Culture

Human gingival fibroblasts were purchased from Provitro AG (HFIB-G, Berlin, Germany). Fibroblasts were maintained in DMEM/Ham’s F12 (Sigma-Aldrich, St. Louis, MO, USA) supplemented with 10% FCS (fetal calf serum; Bovine Calf Serum Iron Supplemented, VWR International, Radnor, PA, USA) and antibiotics (100 U/mL penicillin and 100 mg/mL streptomycin) at 37 °C in 5% CO_2_. Human umbilical vein endothelial cells (HUVECs) were isolated from human umbilical cord veins, as described earlier [31]. In short, HUVECs were retrieved by Collagenase H digestion (Roche Diagnostics, Mannheim, Germany) of perfused human umbilical veins and seeded into cell culture plates with a growth area of 25 cm^2^ coated with 1% gelatin. HUVECs were cultured in specialized medium for endothelial cells (Endopan 3, PAN-Biotech GmbH, Aidenbach, Germany) and antibiotics. Cells were not used for experiments for longer than passage 10. Their typical morphology was closely monitored to exclude contamination and undesired effects due to cellular senescence. Tests to exclude mycoplasma contamination (Venor^®^ GeM Classic, Minerva Biolabs GmbH, Berlin, Germany) were carried out regularly.

### 2.4. Cell Viability Assay

For the analysis of cellular viability after treatment with ZnO NPs, we assessed the cellular metabolic activity (CMA) of the cells with the alamarBlue™ Cell Viability Reagent (Thermo Fisher Scientific Inc., Waltham, MA, USA). For this purpose, 5.000 HUVECs and 10.000 fibroblasts per well were seeded in a 96-well plate and cultivated overnight for adherence. The next day, cells received fresh cell culture medium and were treated with the indicated amounts of NPs for 4 h via a serial dilution of the freshly prepared NP stock solution.

After 4 h, the treatment was stopped by replacing the cell culture medium with NPs with fresh medium. Treatment with equivalent amounts of water served as positive control (=untreated) and treatment with 70% ethanol for 10 min as negative control (=dead cells). Treatment with equivalent amounts of free zinc ions in the form of ZnCl_2_, which is completely dissolved in water, served as control. ZnCl_2_ served as control to assess the effects of free zinc ions in comparison to the effects of the nanoparticles. Thus, if 15 µg/mL ZnCl_2_ are mentioned, this means that a zinc concentration was used that corresponded to the amount of zinc in 15 µg/mL ZnO NPs.

The next day, 20 h after the beginning of the treatment, cell culture medium in all wells was replaced by medium containing 10% alamarBlue™, and the samples were incubated for additional 4 h at 37 °C. The results were obtained using a plate reader (Fluoroskan Ascent Microplate reader, Thermo Fisher Scientific, Waltham, MA, USA; ex: 540 nm, em: 600 nm) and normalized to control cells (=100% viability).

### 2.5. Analysis of Cell Death by Flow Cytometry

For flow cytometric analysis, cells were seeded in culture dishes with a growth area of 25 cm^2^. After treatment with the indicated amount of ZnO NPs or ZnCl_2_, cells were harvested with Accutase^®^ solution (Sigma-Aldrich, St. Louis, MO, USA) 24 h after treatment. Apoptosis and necrosis were assessed by staining with propidium iodide (PI, Thermo Fisher Scientific, Waltham, MA, USA) and AnnexinV-FITC (BioLegend™, San Diego, CA, USA) according to manufacturers’ specifications. Cell samples were then analyzed with a BD FACS Canto II flow cytometer (Becton Dickinson, Franklin Lakes, NJ, USA), and data were analyzed via BD FACSDiva™ Software (Becton Dickinson, Franklin Lakes, NJ, USA). Cells without staining were considered alive/vital, while cells that were single-stained by PI necrotic, those single-stained by AnnexinV-FITC apoptotic, and those that were stained by PI and AnnexinV-FITC were considered dead.

### 2.6. Chicken Chorioallantoic Membrane Assay (CAM-Assay)

Previous studies already proved that the chicken chorioallantoic membrane assay (CAM assay) is well suited for assessing the biocompatibility of nanomaterials [32], their fate within the organism, and hemocompatibility, as well as potential toxic effects [33]. Furthermore, the CAM is an excellent platform with which to study angiogenesis [34] and the angiogenetic activity of biomaterials [35,36].

Hens’ eggs (Leghorn) were stored horizontally in an incubator (Brutmaschinen-Janeschitz GmbH, Hammelburg, Germany) at 37.5 °C for 3 days. On day 3 of egg development (EDD 3), 5–6 mL of albumin was removed with a sterile 10 mL syringe and a 21 G × 1 1/2 needle (0.8 × 40 mm) (BD MicrolanceTM Becton Dickinson GmbH, Heidelberg, Germany) from the blunt end. After albumin removal, the eggshell was opened at the top with autoclaved scissors and subsequently covered with ParafilmVR (Sigma-Aldrich, St. Louis, MO, USA) to prevent evaporation. On day 8 of egg development (EDD 8), gelatin sponge slices of approximately 1 cm × 0.5 cm × 0.2 cm were cut from Gelaspon^®^ strips (4.0 × 1.0 × 1.0 cm, Bausch & Lomb, Rochester, NY, USA) and placed onto the CAM following the gelatin sponge–chorioallantoic membrane assay methodology [37]. Before application of the NPs, the gelatin sponges were allowed to moisten and to adhere to the CAM. The NP stock solution was freshly prepared as described above and diluted to the indicated concentrations with high purity water. Then 20 µL of nanoparticle dispersion with the indicated NP concentrations was applied to the gelatin sponge of each egg. Treatment with high-purity water served as control. Immediately after treatment (EDD 8 = day 0), and after 48 h (EDD 10 = day 2), 96 h (EDD 12 = day 4), and 144 h (EDD 14 = day 6) in vivo fluorescence microscopy was performed, placing the eggs horizontally under a microscope (Olympus BXFM, OLYMPUS DEUTSCHLAND GmbH, Hamburg, Germany). Making use of the autofluorescence of the gelatin sponge, four regions of interest (ROI), 1.000 µm × 1.000 µm in size, were placed next to the sponge on day 0 using the software package cellSens Dimension (Olympus Deutschland GmbH, Hamburg, Germany).

The vessels visible inside each ROI were marked manually, and the software calculated the length of all vessels marked. The analysis of the vessel density was carried out by one person to avoid interobserver variability. The plausibility of manual assignment of blood vessels was cross-checked with the online analysis tool “CAM Assay v.2.0.1” for the quantification of blood vessels on chick chorioallantoic membrane assay (CAM assay) (IKOSA^®^ KML Vision GmbH, Graz, Austria). The overall length of all vessels was computed by summation. Then the vessel density per µm^2^ was calculated by dividing the overall vessel length by the summed area of all four ROIs. The ROIs were monitored over time to track the development of the vessel density. Furthermore, the number of branches and junctions per ROI was calculated for each ROI over the course of time, making use of ImageJ open-source software (ImageJ 1.53c, Wayne Rasband, National Institutes of Health, Bethesda, MD, USA, Java 1.8.0_0172, Fiji distribution [38]). To calculate the number of branches and junctions in the vascular tree, the free ImageJ plugin AnalyzeSkeleton that allows analyzing 2D or 3D skeleton images was used (GNU General Public License v3+, Release: 3.4.2) [39].

Vessel densities as well as number of junctions and branches per egg were shown as percentage values in relation to their number on day 0 (=100%), calculated for each egg individually to depict the egg-specific development.

### 2.7. Statistical Analysis

Unless stated otherwise, results are expressed as mean +/− standard deviation (SD). ANOVA tests were used to compare between different groups. A *p*-value of less than 0.05 was considered to be statistically significant. All statistical analyses were performed using Prism 6.01 (GraphPad Software Inc., La Jolla, CA, USA). Statistical significance is denoted as * *p* < 0.05, ** *p* < 0.01, and *** *p* < 0.001.

## 3. Results

### 3.1. Antibacterial Properties of ZnO NPs

First, we wanted to address the question whether the obtained ZnO NPs were able to exert antibacterial effects on *Escherichia coli*. To do so, *E. coli* were cultured in liquid cultures and then exposed to different amounts of ZnO NPs. As depicted in Figure 1, ZnO NPs had bacteriostatic properties. With increasing concentrations, the inhibition of bacterial growth was more pronounced. Concentrations of 200 and 1.000 µg/mL ZnO NPs nearly completely prevented bacterial growth. Considering bacterial growth after 4 h of incubation with the nanoparticles (Figure 1B), concentrations above 50 µg/mL ZnO NPs were able to significantly inhibit bacterial growth.

### 3.2. Cellular Metabolic Activity of Fibroblasts and Endothelial Cells in the Presence of ZnO NPs

To assess the influence of ZnO NPs on two cell types relevant for wound healing, fibroblasts and endothelial cells (HUVECs) were exposed to ZnO NPs for 4 h, and their cellular metabolic activity was analyzed. Treatment with ZnCl_2_ was performed in parallel to exert the effect of free zinc ions, as ZnCl_2_ is completely dissolved in water. We were able to show that 1 µg/mL as well as 5 µg/mL ZnO NPs and also the equivalent amounts of free zinc ions in the form of ZnCl_2_ were able to stimulate the metabolic activity of fibroblasts and endothelial cells significantly (*p* ≤ 0.001).

As shown in Figure 2, ZnO NPs as well as equivalent amounts of free zinc ions in the form of ZnCl_2_ showed similar stimulating effects, with ZnO NPs tending to be more effective in the stimulation of the metabolic activity in fibroblasts. Concentrations of 7.5 µg/mL ZnO NPs and ZnCl_2_ were still able the stimulate fibroblasts (Figure 2A). At concentrations of 15 µg/mL, treatment with ZnO NPs was more or less neutral, while treatment with ZnCl_2_ had negative effects, as was the case with higher dosages of both substances (data not shown). In endothelial cells, we observed stimulating effects of zinc until concentrations of 15 µg/mL (Figure 2B); at 20 µg/mL, treatment with ZnCl_2_ was still more or less neutral, while treatment with 20 µg/mL ZnO NPs began to have negative effects, as all higher concentrations of both substances had (data not shown).

To confirm that the observed increase in cellular metabolic activity of fibroblasts and endothelial cells after treatment with ZnO NPs originated from an enhancement in cell viability and not from the induction of cell death in the further course of time, we analyzed necrotic and apoptotic cell death after treatment with 1, 5, and 15 µg/mL ZnO NPs or ZnCl_2_.

Figure 3 shows that neither treatment with 1 µg/mL ZnO NPs or ZnCl_2_ nor treatment with 5 µg/mL ZnO NPs or ZnCl_2_ resulted in a decrease in the percentage of living cells in the population of endothelial cells and fibroblasts after 24 h. The ratio between apoptotic, necrotic, dead, and living cells basically also remained unchanged 48 h after treatment (data not shown). Untreated fibroblasts exhibited around 95% living cells in the cell population, as fibroblasts also did after treatment with 1 or 5 µg/mL ZnO NPs or ZnCl_2_, respectively. Endothelial cells exhibited about 90% living cells in the cell population, as also did endothelial cells after treatment with 1, 5, and 15 µg/mL. Only the treatment of fibroblasts with 15 µg/mL ZnO NPs led to a decrease in the percentage of living cells in the cell population to around 80% and an increase in apoptotic and necrotic cells compared to the untreated control population. These findings are in line with the results of the cell viability assay.

### 3.3. Biocompatibility of ZnO NPs Assessed in the Chorioallantoic Membrane Assay (CAM Assay)

The biocompatibility of ZnO NPs was assessed in the chorioallantoic membrane assay (CAM assay). The CAM is the extraembryonic, highly vascularized membrane of the chicken embryo that allows for the assessment of biocompatibility and the interaction of applied substances with the vascular system. To do so, the gelatin sponge method was used. In this method, a gelatin sponge is placed on the CAM and the substance to be tested is applied onto that sponge, in our case the ZnO NPs.

In Figure 4, representative images of the gelatin sponge on the CAM and its gradual degradation over time are shown. Images were taken on the day the gelatin sponge and the nanoparticles were applied onto the CAM (day 0 = EDD 8) and after two days (day 2 = EDD 10), four days (day 4 = EDD 12), and six days (day 6 = EDD 14). It can be observed how the development of the vascular system is progressing; the vascular network is branching out more and more, and larger vessels are also being created. The first row of Figure 4 shows the representative development of the CAM of one egg that served as control, i.e., that was treated with H_2_O as control substance. In the second row, a representative development of one egg that was treated with 20 µL of a 100 µg/mL ZnO NPs dispersion is illustrated. It can be seen that the development of the vascular system is comparable in both groups and that it is not disturbed by the ZnO NPs. Thus, the nanoparticles seem to be biocompatible in the concentrations under investigation, ranging from 1 µg/mL, over 10 µg/mL to 100 µg/mL ZnO NPs.

To quantify the development of the vascular system next to the gelatin sponge on which the ZnO NPs were applied, four regions of interest (ROIs) were placed next to the sponge. These ROIs were monitored over time, and the vessels were counted in these regions (red marks in Figure 5D). The number of branches, junctions, and the vessel density were calculated and expressed in reference to the numbers of each individual egg on day 0 in percentage values (Figure 5). This ensured tracking of the development of each individual egg. It facilitated the comparison between different eggs and the normalization of the different starting conditions due to slightly different development speeds of the chicken embryos. In Figure 5 there is shown the development of the vascular system of the CAM following treatment with 1 µg/mL, 10 µg/mL, or 100 µg/mL ZnO NPs in comparison to control eggs expressed as the relative number of branches (Figure 5A), junctions (Figure 5B), and the vessel density (Figure 5C) on day 2, day 4, and day 6 after treatment. It can be seen that branches, junctions, and vessel density increased over time in all treatment groups. Furthermore, it is noticeable that the interindividual differences between the chicken embryos are quite pronounced in all treatment groups. Possibly 10 µg/mL ZnO NPs had slightly negative effects on the number of branches, junctions, and the vessel density on day 4 and day 6 after the beginning of treatment. However, no significant differences between the treatment groups were found.

## 4. Discussion

The demand for biomaterials for tissue engineering is continuously rising. ZnO NPs can be incorporated in a huge variety of materials, and different ZnO-based hybrid materials were already investigated concerning their regenerative properties. However, on a cellular level, there is less known about the effects of ZnO NPs as pure substance—a gap that we wanted to close in the scope of this study. Furthermore, we made use of the CAM assay to study the in vivo biocompatibility, tissue reactions, and the influence of ZnO NPs on angiogenesis. To sum up, we could show that ZnO NPs possess bacteriostatic properties and that they could enhance the cellular metabolic activity of two cell types relevant for wound healing, i.e., fibroblasts and endothelial cells, in addition to observing a wide range of biocompatibility of ZnO NPs in the CAM assay.

The ZnO NPs under study were able to suppress bacterial growth of *E. coli* bacteria in liquid cultures, showing bacteriostatic properties. Following Gudkov et al., we chose *E. coli* as a model bacterial species, as it is the most frequently studied bacterial species in connection with ZnO NPs [13]. We wanted to verify whether the ZnO NPs under study in our experiments possess the antibacterial properties that were described for other ZnO NPs, as these properties depend on their exact size, surface texture, morphology, etc. [40]. The antibacterial properties are attributed to different mechanisms involving the generation of ROS, cell wall damage due to surface defects on the abrasive surface texture of ZnO NPs, membrane permeability, mitochondrial damage, internalization of NPs due to loss of proton motive force, and uptake of toxic dissolved zinc ions [41]. The here-found antibacterial properties of ZnO NPs are in line with previous studies [10,11,12,13].

Up until now, ZnO NPs mostly proved biocompatibility in vitro by lack of cytotoxicity, as shown in viability or LDH assays [42,43]. We here show for the first time that cellular metabolic activity in fibroblasts as well as in endothelial cells can be stimulated by ZnO NPs in a similar concentration range. Thus, two cell types relevant for wound healing can be promoted in their viability simultaneously using ZnO NPs. For endothelial cells, these findings are in line with findings of Barui et al., who also described an increase in cellular viability in HUVECs by treatment with ZnO nanoflowers [44]. Additionally, the stimulatory influence of ZnO NPs on fibroblasts were previously described by Augustine et al. [23,45], even though the pure substance alone was not investigated in this study. Treatment with ZnCl_2_ was performed to obtain an indication of the mechanism underlying the observed effects. Thus, we more precisely asked whether free zinc ions were responsible for the observed effects or whether the ZnO NPs themselves interacted with the cells under study. We were able to show that both—free zinc ions in the form of ZnCl_2_ as well as ZnO NPs—were able to stimulate fibroblasts and endothelial cells. It therefore stands to reason that at least part of the effect of ZnO NPs is attributable to the free zinc ions released from the nanoparticles. This points to an important role of the release of free zinc ions independent of where they stem from. Nanoparticles, however, have the charm that they allow zinc ions to be released gradually over a longer period of time, and they can be easily integrated into other materials. To exclude that the observed increase in metabolic activity was the indication of incipient cell death, we also analyzed apoptosis and necrosis 24 h after treatment with 1, 5, and 15 µg/mL ZnO NPs and ZnCl_2_. We could observe that in the concentration range in which we saw an increase in metabolic activity there was no significant increase in apoptotic, necrotic, or dead cells detectable compared with untreated control cells.

Previous studies described pro-angiogenic [22,23,44,45,46] as well as anti-angiogenic [47,48,49,50] properties for ZnO NPs, indicating that the corresponding characteristics are strongly dependent on the type of nanoparticles and the environmental conditions. The importance of mild reactive oxygen species generation for the formation of blood vessels is well known [51]. ZnO NPs promote the generation of ROS, a property that was proposed to also be involved in their proangiogenic action [46]. Both zinc oxide as well as zinc peroxide share the properties to (I) release zinc ions and (II) be involved in the generation of ROS. Zinc peroxide is the more oxidative substance, which produces oxygen by reaction with water. Zinc oxide, on the other hand, generates superoxide ions, which either spontaneously or in the presence of hydrogen ions are converted to H_2_O_2_ [22,52]. In addition, it is suspected that ZnO NPs as well as ZnO_2_ NPs may have a twofold activity by also contributing to intracellular ROS formation through different mechanisms also involving free zinc ions [53,54]. Based on these properties, both ZnO_2_ and ZnO nanoparticles have already been associated with biomedical applications on the one hand for use against tumor cells [55,56] and on the other hand as pro-angiogenic agents [22] and as an antibacterial substance [13,57,58]. As a potential pro-angiogenic mechanism of action, it is thinkable that the generation of ROS activates the MAPK pathway and induces the release of pro-angiogenic VEGF, as previously described by us with regard to the response to irradiation [59]. It was already shown that zinc can enhance FGF-2-stimulated VEGF release resulting from upregulating activation of p44/p42 MAP kinase [60], and also Augustine et al. found FGF-2 and VEGF being upregulated in response to ZnO NPs [23].

Using the CAM assay, which is excellently suited to study biocompatibility and angiogenesis, we did not find that ZnO NPs had significant influence on the vascular system, neither in positive nor in a negative fashion. We made use of the pure nanoparticulate substance in contrast to other studies [23], which directly incorporated ZnO NPs into a carrier material. We chose this setting to assess the sole effect of ZnO NPs in a wide concentration range from 1 µg/ml to 100 µg/mL independent of the transplant or implant material that is later chosen for the potential clinical application. ZnO NPs can be embedded in a variety of materials, among them gelatin [47], cellulose [22,61], alginate [20,62,63,64], silica gels [65], chitosan [22,62,64,66,67,68,69,70], different hydrogels [20,21,22,67,71,72], polycaprolactone [45,70,73,74,75,76,77], polyvinyl alcohol [63,66], poly (vinlidene fluoride-trifluoroethylene) [45], polyurethane [62], calcium phosphate [78], polymethylmethacrylate [79], and polylactic acid [80], demonstrating the huge spectrum of biomaterials that can be modified using ZnO NPs. Since the CAM assay is an experiment that is not carried out with genetically identical organisms and during the highly dynamic embryonic development, it is a system that is subject to large interindividual variations. However, these interindividual variations are also seen in a clinical setting. In this system, we could not find any pro-angiogenic nor anti-angiogenic effects of the ZnO NPs. However, we were able to show that the nanoparticles showed a high degree of biocompatibility in the CAM assay, even with a long observation period of six days and a wide concentration spectrum.

Furthermore, the CAM assay revealed that ZnO NP concentrations that proved to be cytotoxic in a 2D cell culture of fibroblasts and endothelial cells could possibly still be used in the organism, as the conditions there are clearly different from the exposure in a cell culture plate. This means that zinc concentrations that have shown antibacterial activity can very well be used in the organism since different cell types are present there, and extracellular matrix as well as extracellular fluids cause different conditions compared with those in cell culture, where especially non-immortalized cells sometimes prove to be very sensitive. The extent to which wound healing in a complex organism can actually be promoted by ZnO NPs must be shown by subsequent experiments that also include more complex experimental models. Furthermore, future studies should reveal the underlying mechanisms that account for the wound healing properties of ZnO NPs with respect to the release of ROS and the possible release of angiogenic factors.

## 5. Conclusions

In summary, we showed that ZnO NPs possess favorable properties to further the regenerative capacity of biomaterials. ZnO NPs possess antibacterial properties, and they can enhance the cellular metabolic activity of fibroblasts and endothelial cells, two cell types relevant for wound healing. In addition, we showed that ZnO NPs possess a good biocompatibility in a complex in vivo system. Thus, these are altogether promising results that suggest that research on ZnO NPs as additive substance in biomaterials to further wound healing should be advanced.

## Figures and Tables

**Figure 1 biomedicines-09-01462-f001:**
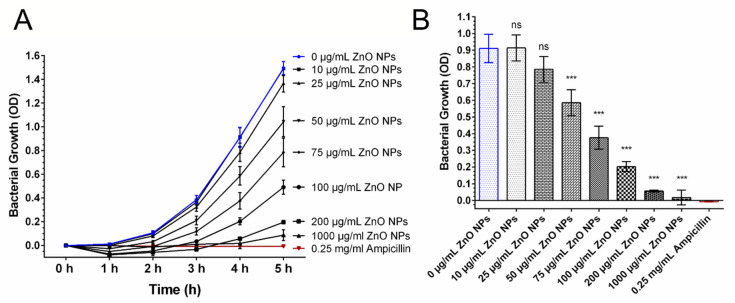
Antibacterial Activity of ZnO NPs. Bacterial growth was monitored by measuring the OD at 600 nm. Bacterial growth was decreased in the presence of ZnO NP (**A**). Within 4 h, concentrations of 50 µg/mL ZnO NPs or higher were able to reduce bacterial growth significantly (*p* < 0.001) compared with untreated bacterial control cultures (**B**). Treatment of bacteria with 10 µg/mL and 25 µg/mL ZnO NPs led to no significant reduction in bacterial growth (*p* > 0.05). Bacteria treated with 0.25 mg/mL Ampicillin served as positive control for antibacterial activity. Shown are mean values ± SD. One-way ANOVA, comparison of each dosage group with untreated control group, correction for multiple comparisons by Bonferroni, N ≥ 3, *** *p* < 0.001, ns = not significant.

**Figure 2 biomedicines-09-01462-f002:**
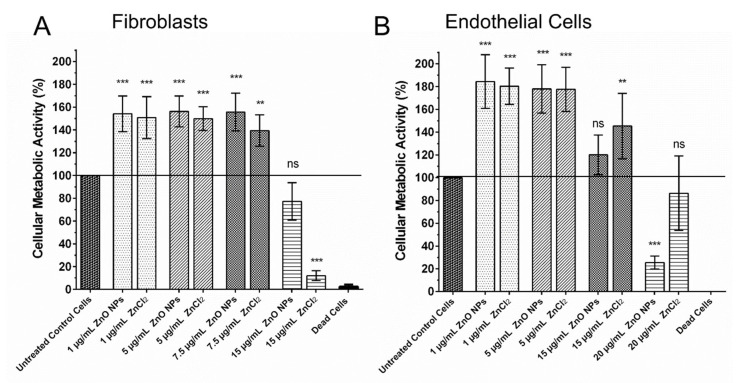
Zinc in the Form of ZnO NPs and ZnCl_2_ can Stimulate Fibroblasts and Endothelial Cells. Treatment of fibroblasts with 1–7.5 µg/mL ZnO NPs or equivalent amounts of free zinc ions in the form of ZnCl_2_ were able to significantly increase their metabolic activity (**A**). Similarly, endothelial cells were significantly stimulated by concentrations of 1–15 µg/mL ZnO NPs or equivalent amounts of free zinc ions in the form of ZnCl_2_ (**B**). Shown are mean values ± SD. One-way ANOVA, comparison of each dosage group with untreated control group, correction for multiple comparisons by Bonferroni, N ≥ 3, *** *p* < 0.001, ** *p* < 0.01, ns = not significant.

**Figure 3 biomedicines-09-01462-f003:**
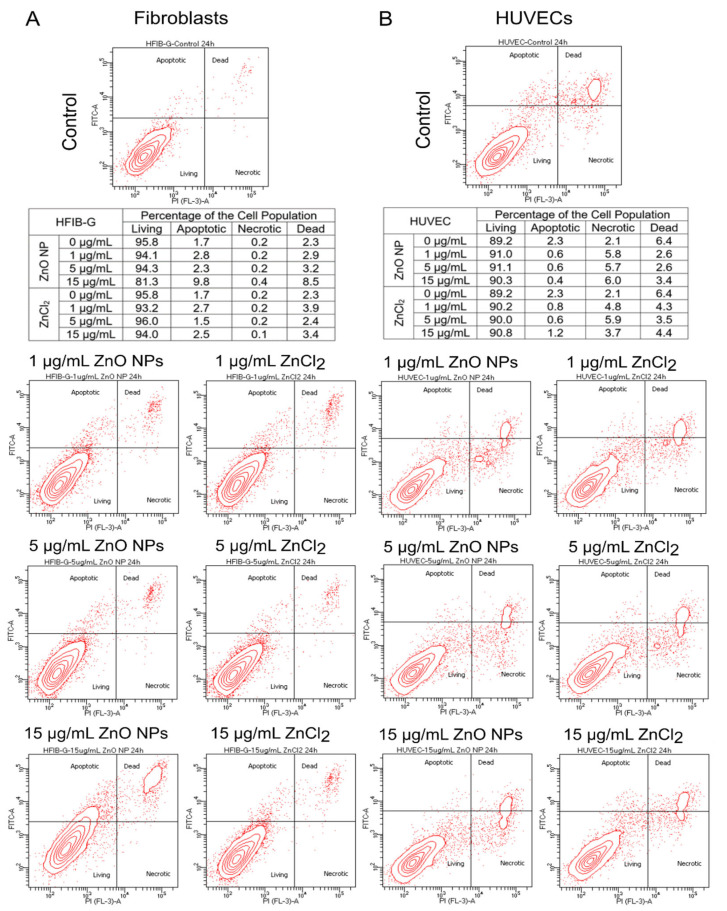
Relative Proportions between Living, Apoptotic, Necrotic, and Dead Cells after Treatment with 1, 5, or 15 µg/mL ZnO NPs or Equivalent Amounts of ZnCl_2_ in Fibroblasts (**A**) and Endothelial Cell Populations (**B**). Whether cell death occurred following treatment with ZnO NPs or ZnCl_2_ was explored by detection of apoptotic, necrotic, and dead cells in the cell population via staining with propidium iodide (PI) and annexinV-FITC and subsequent flow cytometric analysis. In (**A**), the results from the fibroblast populations are shown, while in (**B**), those from the endothelial cell populations are shown. The contour blot at the top of each column shows the proportions between living, apoptotic, necrotic, and dead cells in the untreated control population. In the lower left quadrant, living cells are located; in the upper left quadrant apoptotic cells can be seen; in the upper right quadrant are dead cells; and in the lower right quadrant are necrotic cells. Underneath in the table, the raw values of the exemplary measurement are shown (N = 1). Below, the two-by-three figures show each treatment group in a contour blot individually, with the fibroblasts on the left-hand side and the endothelial cells on the right-hand side. Treatment with 1 or 5 µg/mL ZnO NPs or equivalent amounts of ZnCl_2_ did not lead to an increase in the proportion of apoptotic, necrotic, or dead cells in the cell populations, neither in fibroblasts nor in endothelial cells. In the range of concentrations studied here, only the treatment of fibroblasts with 15 µg/mL ZnO NPs led to an increase in the percentage of apoptotic and dead cells in the cell population.

**Figure 4 biomedicines-09-01462-f004:**
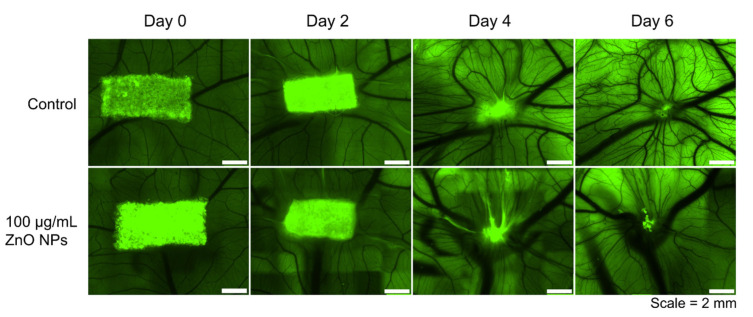
Representative Images of the Development of the Chorioallantoic Membrane treated with ZnO NPs Applied onto a Gelatin Sponge. The vasculature of the chorioallantoic membrane developed rapidly from EDD 8 (day 0) to EDD 14 (day 6) (first row). Treatment with 100 µg/mL ZnO NPs onto an autofluorescent gelatin sponge (in green) did not influence the development (second row). The gelatin sponge is gradually degraded by the highly vascularized CAM over time. There were no significant negative tissue reactions observed following treatment with ZnO NPs.

**Figure 5 biomedicines-09-01462-f005:**
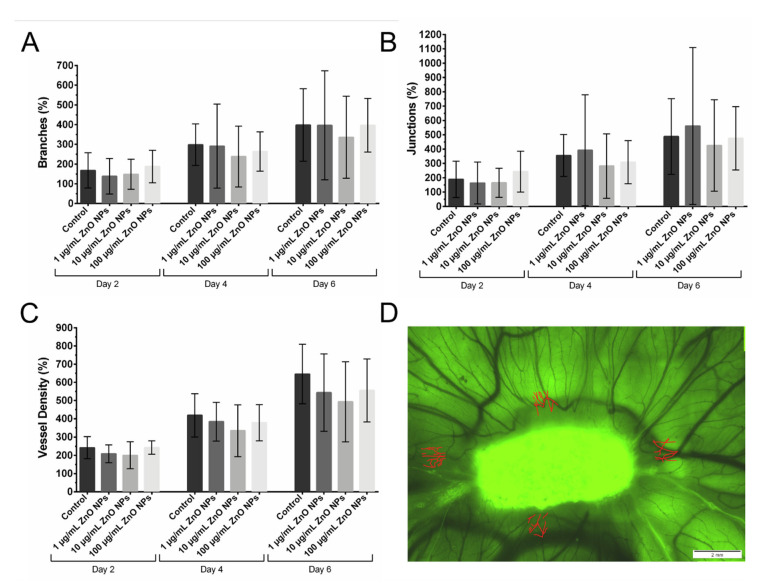
Vascularization of the CAM after Exposure to ZnO NPs. The vascularization of the CAM developed rapidly from EDD 8 (day 0) to EDD 14 (day 6). The number of branches (**A**), junctions (**B**), and the vessel density (**C**) increased in the period under investigation in all dosage groups. All three parameters are expressed in relation to the number of branches, junctions, and vessels on day 0 (=100%) of each individual egg to account for individual development of each chicken embryo. In (**D**) there is shown a representative image of an egg on day 2. The gelatin sponge (in green) is beginning to be degraded, and in four ROIs next to the sponge, the vascularization was quantified. In red, the vessels detected in the ROIs can be seen. Then the number of branches, junctions, and the vessel density in all ROIs were calculated. Shown are mean values ± SD. Two-way ANOVA, comparison of each dosage group with untreated control group at the specific day, correction for multiple comparisons by Bonferroni, N = 3 (three independent runs) with 106 eggs in total being included in the analysis; eggs per dosage group ≥ 14. No significant differences were found between the different dosage groups.

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
