# Peer review of "Zinc Oxide Nanoparticles Exhibit Favorable Properties to Promote Tissue Integration of Biomaterials"

_biomedicines, 2021, doi:10.3390/biomedicines9101462_

Round 1

Reviewer 1 Report

Dear Editor,

The manuscript (biomedicines-1379289) aims to study and report the beneficial effect of ZnO nanoparticles on biocompatibility, angiogenesis, and wound healing. The paper is well-written and is organized. Yet, considering the fame of this material for biomedical applications and a long history of study on it alone or as composite, as widely found in the literature, I was wondering what is the novelty in this study? A new perspective, new characterizations leading to uncovering new mechanisms? Implementation of the CAM assay as a new testing approach for the ZnO nanoparticles alone is not convincing and does not impress the reader to look at this work as a novel study. My other major concerns include:

1) Page 5; "4 hours" is long enough to validate the positive effect of the ZnO nanoparticles on cellular metabolism?

2) Discussion- 1st line; why smart?

3) why E. Coli has been based for the antibacterial test?

4) Considering that ZnO produces ROS that later trigger further vascularization through enhanced VEGF release, why the authors have not quantified the concentration of ROS in the medium?

5) What kind of wound the authors mean in their study? If the skin wound healing is a target application for the ZnO nanoparticles? Keratinocyte alongside fibroblast matter the most.

6) There is a study in the literature (Nature communications 8 (1), 1-10, 2017) that proposes ZnO2 as a cytotoxic material with potential anticancer applications. What would be the difference in operation mechanism between ZnO and ZnO2 and the ROS release? Please cite this work and discuss the likely discrepancies in terms of biological effects.

Author Response

Reply to Reviewer I

[…] Yet, considering the fame of this material for biomedical applications and a long history of study on it alone or as composite, as widely found in the literature, I was wondering what is the novelty in this study? A new perspective, new characterizations leading to uncovering new mechanisms? Implementation of the CAM assay as a new testing approach for the ZnO nanoparticles alone is not convincing and does not impress the reader to look at this work as a novel study. […]

Thank you for this comment which has shown us that we have not yet been able to sufficiently explain the rationale behind the experiments performed. Three important points of this study are new, and were little studied so far: first, we use healthy, primary cells in contrast to most studies that describe effects on tumor cells. These cells better represent the conditions in the body as they are not immortalized in contrast to tumor cell lines. Second, we deal with very low ZnO NP concentrations, and last but not least, we study the wound healing capacities of ZnO NPs as a single substance not directly integrated into other materials. All these points are important to understand the fundamentals of the effects these nanoparticles can have in a wound healing setting. Of course, only a small aspect of wound healing can be considered here. We are aware of this limitation and further studies can certainly describe the overall picture even better. To better describe the rationale behind our study, we included some words on the new aspects that were studied in the introduction (ll. 86).

[…] 1) Page 5; "4 hours" is long enough to validate the positive effect of the ZnO nanoparticles on cellular metabolism? […]

Thank you very much for bringing up that point. In fact, an equilibrium between the treated cells and the nanoparticles is established quite quickly in less than 4 h in cell culture systems. The extracellular release of zinc ions from ZnO NPs is initially quite rapid, but then slows down as the proteins in the cell culture medium form a protein corona around the nanoparticles and the nanoparticles agglomerate. We also observed that nanoparticles are present on the cells and in the cells after 4 h. All this leads to the fact that 4 hours are long enough to validate the positive effects of the nanoparticles. Cell stimulation or cell death after treatment takes longer, of course, because nanoparticles may also release zinc ions intracellularly. Therefore, we change the medium after 4 h, but the effects of the treatment are determined after 24 h.

[…] 2) Discussion- 1st line; why smart? […]

Thank you for that comment. In order to avoid misunderstandings, we deleted the word “smart” (l.360).

[…] 3) why E. Coli has been based for the antibacterial test? […]

We used E. coli as a model bacterial strain as it is the most frequently studied microorganism in connection with ZnO NPs. We included this aspect also in our manuscript to make it clear for the reader (ll. 370).

[…] 4) Considering that ZnO produces ROS that later trigger further vascularization through enhanced VEGF release, why the authors have not quantified the concentration of ROS in the medium? […]

Thank you very much for bringing up this aspect. The main focus of the present publication was on the functional effects that ZnO NPs have, not on the underlying mechanisms. In addition to studying the amount of ROS released both extracellularly and intracellularly, it would certainly be highly interesting to study the release of pro-angiogenic factors such as VEGF and FGF after treatment to understand the exact mechanism of action. Furthermore, studying the MAPK signaling pathway could reveal what exactly happens within the cells after treatment. We totally agree with you, in any case, these aspects are highly interesting and will be the subject of future studies. We included that aspect also in the outlook of the discussion (ll. 450).

[…] 5) What kind of wound the authors mean in their study? If the skin wound healing is a target application for the ZnO nanoparticles? Keratinocyte alongside fibroblast matter the most. […]

Thank you very much for drawing our attention to that aspect. Indeed, we primarily imagined not a skin wound as a treatment environment for the ZnO NPs but rather tissues within the body such as the interface of dental implants, orthopedic implants, and other transplants and implants that are inserted into human tissues. In this setting, to our knowledge fibroblasts which often adhere first to the foreign material and endothelial cells besides the cells of the immune system are the most decisive cells for successful tissue integration. To make our choice of cell types more transparent to the readers of the study we included a paragraph on that topic in the introduction (ll. 66).

[…] 6) There is a study in the literature (Nature communications 8 (1), 1-10, 2017) that proposes ZnO2 as a cytotoxic material with potential anticancer applications. What would be the difference in operation mechanism between ZnO and ZnO2 and the ROS release? Please cite this work and discuss the likely discrepancies in terms of biological effects. […]

Thank you very much for that comment. We totally agree with you, that besides ZnO NPs also ZnO2 NPs are of high interest for different biomedical applications especially when the release of ROS is desired. This is the case for biomaterials used to exert pro-angiogenic properties in a wound healing setting and as well in a cancer therapy setting where the generation of ROS in tumor cells could contribute to tumor cell death. In general, ZnO2 is the more oxidative substance compared to ZnO and the mechanism of ROS generation is slightly different. Thus, the potential of ZnO2 NPs to generate ROS is higher, however, studies on its biocompatibility are more sparse compared to ZnO NPs. All in all, both nanoparticle types have interesting properties – also but not only concerning ROS generation – and both of them should be pursued further. As proposed by you, we also included these aspects in our discussion (ll. 410).

Reviewer 2 Report

The authors aim to investigate the biocompatibility of ZnO nanoparticles. Data show that at high concentrations suspensions of ZnO nanoparticles can inhibit the growth of E.coli. Nanoparticles as well as ZnCl2 solution activate the metabolism of fibroblasts and endothelial cells at concentrations below 15µg/ml. Application of ZnO nanoparticles did not show changes in angiogenesis in a CAM assay. The authors conclude that

Unfortunately, some major issues arise when reading the manuscript.

The title states that ZnO nanoparticles could be incorporated into biomaterials for tissue regeneration and thus could improve the healing process. However, ZnO nanoparticles were not incorporated into biomaterials and tissue regeneration was not investigated. Please choose another title that better fits the data shown and change the introduction accordingly.

The authors claim that nanoparticles show bacteriostatic properties, enhance metabolic activity of relevant cell types and show biocompatibility in the CAM assay. Yet, they use different particle concentrations for their underlying experiments.

  • Bacterial growth is only inhibited at concentrations above 50µg/ml.
    It seems relevant human cells are more sensitive to ZnO than bacteria.
  • Fibroblasts and endothelial cells show toxicity at or above 15µg/ml. Increased metabolic activity and ROS production can precede cell death and is often not a sign for increased viability and regenerative potential. Since this seems to be the major claim it should be confirmed by complementary methods and results should be discussed accordingly.
  • In contrast to other test substances regularly used in the CAM assay, applied nanoparticles are only in contact with the CAM after degradation of the sponge or the nanoparticles. Since ZnO has previously been shown to induce angiogenesis, the data contradict previous studies and should be discussed in more detail.

No data to characterize the particles (size distribution, degradation, purity, …) are included in the manuscript.

Minor issues:

Is this the definition for how vessel density was calculated in the manuscript?

“The overall length of all vessels was computed by summation and the vessel density per μm2 was calculated by dividing the overall vessel length by the summed area of all four ROIs.”

If it is, please state it clearly. To confirm manual assignments use more than one operator.

If a software plugin is used to quantify branch points and junctions please state the plugin name and version.

What are the red annotations in Figure 4D?

Please check References again! E.g. References are missing for the first sentence of the discussion. Reference 8 in the Discussion section seems misplaced. References for ROS activation and angiogenesis are missing yet biomaterials are excessively referenced…

The manuscript could be improved in terms of style and grammar. Please also check for missing words. E.g. on page 5 paragraph 3.2 the last sentence is missing the word nanoparticles. The figure legend of Figure 1 is missing the word Ampicillin.

Author Response

Reply to Reviewer II

[…] The title states that ZnO nanoparticles could be incorporated into biomaterials for tissue regeneration and thus could improve the healing process. However, ZnO nanoparticles were not incorporated into biomaterials and tissue regeneration was not investigated. Please choose another title that better fits the data shown and change the introduction accordingly. […]

Thank you very much for your comment on the title of our study. We changed the title to “Zinc Oxide Nanoparticles Exhibit Favourable Properties to Promote Tissue Integration of Biomaterials” (l. 1-3)

[…] The authors claim that nanoparticles show bacteriostatic properties, enhance metabolic activity of relevant cell types and show biocompatibility in the CAM assay. Yet, they use different particle concentrations for their underlying experiments.

  • Bacterial growth is only inhibited at concentrations above 50µg/ml.
    It seems relevant human cells are more sensitive to ZnO than bacteria.
  • Fibroblasts and endothelial cells show toxicity at or above 15µg/ml. Increased metabolic activity and ROS production can precede cell death and is often not a sign for increased viability and regenerative potential. Since this seems to be the major claim it should be confirmed by complementary methods and results should be discussed accordingly. […]

Thank you very much for drawing our attention to that aspect of our study. You are totally right, that the concentrations that exhibited bacteriostatic properties and those which stimulated fibroblasts and endothelial cells diverged – at least in the cell culture plate. Taking into account that the environmental conditions in an organism with extracellular matrix and extracellular fluids being present we are quite hopeful that in such a setting higher ZnO NP concentrations can be used. Having in mind that concentrations up to 100 µg/ml ZnO NPs did not show any negative effects in the CAM assay this hope is supported. Of course, this should be further investigated in studies using more complex experimental model systems. As we think that the context of the results is also important for the readers of the study, we have added a paragraph in the discussion that deals with these aspects (ll. 447).

The second very important aspect that you brought up, is that increased metabolic activity can precede cell death and is often not a sign of increased viability and regenerative potential. We followed your suggestion to confirm our claim that ZnO NPs are able to stimulate fibroblasts and endothelial cells in a genuinely positive way by performing an apoptosis-necrosis assay via flow cytometry. This assay indeed validated that no cell death occurred at the concentrations that we found to stimulate fibroblasts and endothelial cells. The experimental procedure of this additional assay was added to the materials and methods section (ll. 162), the results are described in ll. 270 and in Figure 3 and they were discussed in ll. 399.

[…] In contrast to other test substances regularly used in the CAM assay, applied nanoparticles are only in contact with the CAM after degradation of the sponge or the nanoparticles. Since ZnO has previously been shown to induce angiogenesis, the data contradict previous studies and should be discussed in more detail. […]

Thank you very much for your comment on that aspect. One problem that arises when substances are applied directly to the CAM is that it is difficult to find the exact spot where the substances were applied to the CAM. The CAM is subject to very rapid growth and profound changes so that after just one day without any kind of marking, you will no longer be able to find where the substances were applied to the CAM. Often rubber rings or cotton balls are used to find the application site again, but in our experience, these vehicles have the disadvantage that they have an irritating effect on the CAM. Therefore, we decided to use gelatine as a marker where the nanoparticles were applied. In fact, the application is not directly onto the CAM, but the gelatine sponge gets time to be moistened on the CAM before the application of the nanoparticles and, moreover, the nanoparticles can easily penetrate the pores of the sponge, which have sizes in the micrometer range, and thus reach the CAM. We fully agree that this application method also has its weaknesses and disadvantages, but it has also already been described frequently, for example in Ribatti, D.; Nico, B.; Vacca, A.; Presta, M. The gelatin sponge-chorioallantoic membrane assay. Nat. Protoc. 2006, 1, 85–91, doi:10.1038/nprot.2006.13. Looking at the literature on zinc oxide nanoparticles, we can see that they have been associated with both pro-angiogenic and anti-angiogenic effects. Therefore, it cannot be said in general that our study contradicts all previous investigations. To clarify these aspects, we have also included the corresponding sources as references in the manuscript (l. 191, 405, 406). Furthermore, we advanced the discussion according to your suggestion (l. 396-421).

[…] No data to characterize the particles (size distribution, degradation, purity, …) are included in the manuscript. […]

We totally agree with you that we did not include any chemical or chemico-physical characterizations of the nanoparticles in the manuscript. As our expertise lies in molecular and cellular biology, we refrained from characterizing the particles ourselves. On the contrary, we used nanoparticles that are commercially available and that are characterized by the company (IoLiTec Ionic Liquids Technologies GmbH, Heilbronn, Germany) that synthesized them and that is specialized to these analyses. All necessary information on the nanoparticles can be found in the materials and methods section (ll. 99).

[…] Is this the definition for how vessel density was calculated in the manuscript? “The overall length of all vessels was computed by summation and the vessel density per μm2 was calculated by dividing the overall vessel length by the summed area of all four ROIs.” If it is, please state it clearly. To confirm manual assignments use more than one operator. […]

Thank you very much for your comment on that aspect. In order to clarify our procedure, we advanced the corresponding section in the materials and methods (ll. 203). We indeed marked the blood vessels, but we validated the administration with the Ikosa software. That information was also added to the manuscript.

[…] If a software plugin is used to quantify branch points and junctions please state the plugin name and version. […]

This information was added to the manuscript, as proposed by you (ll. 213).

[…] What are the red annotations in Figure 4D? […]

Thank you very much for pointing out that this part of the caption was missing until now. We corrected that (ll. 353).

[…] Please check References again! E.g. References are missing for the first sentence of the discussion. Reference 8 in the Discussion section seems misplaced. References for ROS activation and angiogenesis are missing yet biomaterials are excessively referenced… […]

Thank you for reading our manuscript so carefully. We apologize for the mistake! The error has been corrected (l. 380). Furthermore, additional references were added in the discussion (l. 405-426).

[…] The manuscript could be improved in terms of style and grammar. Please also check for missing words. E.g. on page 5 paragraph 3.2 the last sentence is missing the word nanoparticles. The figure legend of Figure 1 is missing the word Ampicillin. […]

The errors mentioned have been corrected and in addition, the manuscript has been proofread in detail by all authors to eliminate further errors

Round 2

Reviewer 1 Report

Dear Editor,

My major comments have been properly addressed.

Reviewer 2 Report

The authors have improved the manuscript and have adressed major questions. Remaining open questions are discussed and directions for future research are proposed.